# The Social and Cultural Dimensions Associated with Death in Muslim Communities, a Case Study Khartoum City

Osman Sirajeldeen Ahmed [1,]*, Elsayed Abdalrahman [2], Alaa Zuhir Al Rawashdeh [1] and Asma Rebhi Al Arab [1]

1   Department of Sociology, Ajman University, Ajman 346, United Arab Emirates
2   Department of Sociology, University of Science and Technology of Fujairah, Fujairah 2202, United Arab Emirates
*   Correspondence: o.ahmed@ajman.ac.ae

**Abstract:** In Arab universities, sociologists rarely discuss the sociology of death. By studying social and cultural variables along with subjective and objective meanings of death, this paper contributes to filling this gap in research on death in a Sudanese urban area. Furthermore, the study examines the relationship between the burial of the dead and the time and place of their burial, social status, relatives, and religious affiliation as they relate to their burial. A major objective of the research is to explore the social and cultural dimensions of death in Sudanese communities. Data were collected using interviews and observations in the field using the descriptive method. Death is more of a social than a biological fact; therefore, the general findings of this research are about declaration of death, and what it implies about social cohesion. Burial and social acts following death are acts that express social meanings, and further, indicate how biological death has occurred. Based on variables such as social status, family relationship, and religious affiliation, it can be seen that the deceased person and/or family holds these beliefs.

**Keywords:** sociology of death; death; FATWA; Beit Al-Bika; Wad Al-Lahad; Khartoum

## 1. Introduction

Until recently, death was not a topic that received a lot of attention from sociologists. Although it does not differ from the other social processes and social attitudes in society, members of society do not interact with it on a daily basis, due to its socially and psychologically unpopular position for society's members. In spite of this, it is still important to study it as an objective part of society structure as well as a case defining life and social reality. In sociology, there is an argument that death has been ignored and has been banished from representing life forms, causing a vacuum in the field of death study. Studying death as a social phenomenon in Arab countries clarifies the picture.

In this study, death is investigated from a socio-cultural perspective because it is regarded as a social status accompanied by a set of social actions that form a social reality. In this study case of Khartoum city urban community, we examine realistic and symbolic cultural and social practices associated with death.

## 2. Research Questions

A key research question in this study remains the relation between realistic societal formations and individuality toward death, as well as the societal projection of death with accompanying cultural perceptions that are supported by the societal death criteria. Thus, the study aims to answer the question of how death is perceived socio-culturally.

## 3. Methodology

Firstly, the research employed a descriptive methodological approach. It can be helpful when relatively little is known about a topic or problem, as in this case. Through the study sample, observations, descriptions, and social anatomy of death, as well as the

accompanying social and cultural actions, are used as part of the analysis method to understand the social act expressed in the interviews, emphasizing social and behavioral aspects to support the cognitive aspects of death studies.

Secondly, in the second step, we will discuss the methods of collecting data.

a.  Our discussion groups were based on interviews we conducted in Beit Al-Bika[1] and selected burial grounds, both with the family of the deceased or with those present for the duty of condolences. This study provides insight into the social and cultural dimensions of death through a combination of diverse religious, cultural, social, economic, and educational backgrounds.

b.  Observations: The researchers used the tool of simple observations to make about the behavior patterns of the family members around the time of death announcement, burial, and days of mourning for the deceased.

Thirdly, the design of the research sample: 175 families whose members died were used as an intentional sample in the study, following the news bulletin broadcast by Omdurman Radio at half past six in the morning and eight at night, which announced the deceased, the funeral location, and mourning arrangements during the period 1–29 August 2019.

A sample of 236 cemeteries was randomly selected from the Khartoum state's 236 cemeteries. These ancient cemeteries include Farouk, the Lamab, Burri Al-Mahas in Khartoum, Safia, Halfaiat Al-Mulook, Sheikh Khojaly and Sheikh Hamad in Khartoum North, and the cemeteries of Lamab, Burri Al-Mahas, Sheikh Khojaly, and Sheikh Hamad in the state of Khartoum. Besides that, a study was conducted of Hamad El-Neel, Ahmed Sharfi, and El-Bakry's tombs in Omdurman. Besides their historical significance and proximity to densely populated areas, they are also symbolic of social and religious values, which is why they represent the city's oldest cemeteries.

## 4. Brief Literature Review

It has been recognized by sociologists that death and funeral behavior are poorly studied in our society, especially in modern social systems, and that sociology has not given death a high priority (Kellehear 2008). Even so, we must note that death studies, especially those focused on their philosophical aspects, have the potential to bring up concerns raised by Western sociology of death, which has assumed the role of comparison and glorification of symbolism (Durkin 2003).

A great deal of social anthropological research was devoted in the twentieth century to death as a rite of passage and the functions it performs in social construction (Walter 2012). Throughout Western thought, death has tended to be perceived as a philosophical concept rather than as an objective reality affected by societal norms (Greek and contemporary philosophy, Heidegger 1976).

Many social and cultural issues concerning death have arisen in the wake of the migration of Arabs and Muslims from the Middle East to Europe in the modern era. By studying topics related to burial sites, confrontations between immigrants and institutions, as well as public opinion on the matter of respecting Islamic burial rituals, burying the dead in the original site, and the socioeconomic impact of immigrants' deaths (Ahmed 2020), several studies have examined the issue of death in immigrants' lives and how the image of the other takes after death.

In the research, (Al-Aqiba 2007), an anthropological study, was one of the Arab studies on which it relied. By describing all such practices in relation to social and economic changes in the Syrian coast region in the event of death, (Al-Aqiba 2007) examined the social meanings of traditions associated with death.

As the 1990s ended and the new millennium began, academics became more interested in examining death's social and cultural aspects.

In his study, Seale (2000) emphasized the differences in gender and socioeconomic status within countries. Additionally, he summarized the current knowledge regarding death pathways, outlined knowledge gaps, and outlined the challenges and dilemmas

associated with providing formal health care to dying people in underdeveloped and developed countries. In addition to discussing traditional religious consolation for death and bereavement, his paper explored traditional religious resources.

An important feature of the social history of the last century is the observed reduction in mortality, as examined by Riley (1983). Social inquiries about death and dying have a promising future, according to the review, which concludes that meanings of death are changing constantly. Among the key questions that remain unanswered are: Will socialization for death become a reality? Is it likely that dying people will strive to maintain a greater sense of independence as they age? Are there fewer ethical dilemmas associated with passive euthanasia? The number of elderly people who commit suicide will continue to increase. Will suicide remain the "last option" for them? Towards the future, will widows have to cope with new patterns of bereavement? Will the terminally ill be institutionalized in new care environments? Will death become more accepted in the future?

Sociological studies have consistently concluded that death is a societal reality which cannot be denied or avoided, and that societal cultures do not perpetuate a death denial at the individual or societal level (Walter 1991; Kerridge et al. 2002; Thompson et al. 2016).

A new concept of death has emerged in the Arab region as a social act resulting from issues related to development, politics, and society, according to Osman Ahmed (2020). This article discusses extrajudicial executions, killing on the basis of identity, suicide, and death associated with jihadism and euthanasia.

## 5. Death in Sociology Studies

A relatively recent research field in science is the study of death, especially in sociology. Studying death from a sociological perspective was unclear until the mid-1950s. A reluctance to think about death may result from social research into this area.

According to social realities surrounding death, the field of death studies developed rapidly in American and British universities beginning in the nineties of the previous century. In American and British universities, such as the University of New York, California and Newcastle University, death studies courses became part of the sociology curriculum under the name sociology of death. Death as a social issue, funerals, religion, death, the media, and death process, final diagnosis, theories of death, and the grief and bereavement process are some of the topics explored within this branch. The theoretical heritage of death, however, cannot be overridden in classical sociological studies, considering the recent interest in studying death in sociology. The separation between the living world and the dead is crucial for social progress, as Herbert Spencer argued. Durkheim (Durkheim 2011) attributed various suicide patterns to social disease based on anomism theory. There has been extensive and diverse research published on suicide in the field of social integration studies. The time of one's death is often influenced by social factors other than suicide, from a sociological perspective. According to Thomas Jefferson and John Adams, some people tend to commit suicide after taking part in large celebrations and events. As Phillips and Feldman note in several empirical studies, "anniversary effects" occur when significant social events precede fewer deaths than expected, suggesting that if some people are so disconnected from their communities that they commit suicide, it is more likely. As a result of Durkheimian thought about suicide (Durkheim 2011), others may delay their death so they can participate in social events of great importance.

This is opposed to the functionalists, who were concerned about issues of social loss and reintegration into society once one had died.

Talcott Parsons is considered one of the most influential figures in the theory of social death. Social theorizing of death in sociology evolved into applied studies, as presented in Marshal & Levey's study on socialization of death, and in contrast to theoretical work, applied studies have become more common.

The applied literature shows that social research related to death were, and are to a large extent, objective.

Death taboos, social causes of death, death planning, hospital social organization of death, and funeral rites are some examples of these studies.

Studies of death have focused on how people manage death in hospitals and how they receive health care for those who die, especially those with diseases that cause death like cancer (Exley 2004; Katz and Peace 2003).

In addition to demographic studies and population, sociology's interest in social death also appears through demographic studies, population sociology, and the study of age and mortality rates. Sociologists have been using social death on a large scale for the last three decades, such as Zygmunt Bauman, who refers to death resulting from apartheid and ethnic cleansing, identity killings, persecution, and genocide (Králová 2015).

Sociology of the body has been focusing on death in relation to the body, particularly with regards to today's modern individual, by studying relationships between the body and death (Chris 2012).

It is through understanding the inevitability of death that Peter Berger understands the social significance of the human body, meaning that it can only be understood through the lens of death as a feature of the human condition. People in his relationship with them receive significant connotations related to the socially problematic reality because the socially problematic reality gives them meaning, but these subjective connotations impose a different objective reality from which humans may experience existential problems regarding themselves. In Berger's view, death is a problem for people because of their limitations in terms of the body (Berger and Luckman 1967). An alternative approach to death in the sociology of the body is found by Anthony Giddens, who, through his analysis of the concept of lifestyle, concluded that modernity has caused individuals to place increasing pressure on death, which they find disturbing since they realize that their own lives are characterized by contradiction and continuity (Giddens 1991).

A recent study by Chris (2012) asserts that modern forms of the body create an existential problem with respect to death as the result of the contemporary problem of death.

There has been an increase in terrorism and the resulting death as a result of political, cultural, social, and economic variables globally, especially after the events of September 2001. Under the name of cultural formations of global politics, Huntington (1997) referred to this as the clash of civilizations.

In sociology, euthanasia and organ donation are also ethical and humanitarian issues that affect the study of death (Al-Nahdi in 2017). Furthermore, Muslim immigrants differ culturally from other peoples, as well as the indigenous population of their countries in which they immigrate, particularly in Europe. There are a variety of differences in the burial practices of deceased individuals and funeral rites, as well as the issues relating to immigration and death resulting from adventure in immigration. It is noteworthy that although this research addresses death from a social perspective, there has not been a dedicated study in sociology on the sociology of death. Alternatively, we may be able to anticipate a promising future for specialization related to death which has not yet been answered, such as: Will socialization of mortality become a reality? How does euthanasia pose ethical dilemmas? Is the trend of elderly people committing suicide likely to continue? How will widows cope with bereavement in the future? Are terminally ill patients likely to be institutionalized in new care environments? Aside from these, there are other questions that have emerged from the social changes in the contemporary world (Riley 1983).

## 6. Social and Cultural Dimensions to the Death in Sudanese Society

Through a focus on Khartoum, especially among the Muslim community, this part of the study explores how death is related to social and cultural norms in Sudanese society. Several topics will be discussed in this context, and it should be noted that the data had been collected using an open discussion tool that is based on interviews as well as simple observation tools.

## 6.1. Social Declaration of Death

There is a social readiness to accept death cases in incurable cases and to support the family of the patient who will die. The death may be announced even before it occurs for incurable cases.

Whenever the death of a person is announced, a state of social transformation occurs. Not only does the announcement convey a biological state, but it also conveys a social situation based upon relational ties, neighborhood relationships, and friendships (sometimes even work colleagues). A funeral tee, funeral tray, supplies, and meal preparation are all part of this formation. It works together with the family to ensure that the deceased is shrouded in a social aspect of death, that burial cemeteries are chosen, that those digging the grave are selected, and that the funeral tray and its supplies are prepared. According to the field interviews conducted with the research sample, announcing the death state in its entirety is not primarily related to the biological aspect, but rather to the social aspect, referred to in Sudanese vernacular as "hiding the case."

A second social transformation characterized by religion can be observed when the deceased is characterized as having a religious status by adding the word "the late" to their name, and it is sometimes only necessary for the deceased to be pronounced among the deceased's family, and this is true in all sects of Sudanese Muslims. According to Sunni beliefs, this name is a socio-religious expression and therefore is not permitted. A fatwa based on Sharia No. 8217 has been issued (Appendix A).

## 6.2. How to Announce Death Publicly?

A death is announced by a number of traditional and modern methods, for example, when a person is close to their neighborhood by hearing the women wail, or in a mosque, whereas when a person is close to relatives, they communicate with them through various means, including WhatsApp, which is currently playing an important role. Besides gatherings around the house of the deceased, other social signs indicate that the death has been announced, such as erecting a funeral hall.

Between half past six in the morning and eight in the evening, national radio broadcasts are a very important way for the community to announce a deceased person's death, which announces the deceased's name, the name of his closest male relatives, as well as the burial location and mourning place. In this regard, historian Muhammad al-Tayeb states Radio Omdurman began broadcasting updates on the deceased in the fifties of the last century. Al-Tayeb (2010) describes how gradually people became familiar with the news of the radio regarding the dead.

It is an indicator of the social aspect of death to announce the dead on the radio, as the matter goes beyond death to social pride. Especially important is mentioning the senior jobs of the deceased's sons, especially those who served in positions of authority. Al-Tayeb (2010) states that the deceased's brothers and cousins are chosen on the basis of their social status.

## 6.3. Burying the Dead

Sudanese society generally does not go through legal and social complications when it comes to burying the dead. A deceased's body is placed in the cemetery after it has been washed and shrouded. It depends on the proximity or distance of the deceased's house to the cemetery on how the funeral will be transported to the cemetery. An excavation of the grave would first be supervised by a group of relatives or neighbors, particularly in old cemeteries in Khartoum.

A funeral cemetery's selection is influenced by several social and cultural considerations: Social position, religious affiliation, and kinship group.

As a first step towards clarifying these considerations, it is useful to identify the geographical, religious, and historical distribution of the cemeteries of Khartoum, the capital, noting that most of these cemeteries are situated in densely populated areas and are located in the prominent locations as follows:

1.  Farouk's tombs in Khartoum's southeastern region, which belonged to educated aristocratic families.
2.  Al-Jarif cemeteries west and north-east of Khartoum.
3.  Burri al-Mahas cemeteries, located in the east of the city of Khartoum near the Blue Nile.
4.  Burri Lamab cemeteries east of the city of Khartoum and were associated with religious affiliation to the Hindi Sufi order.
5.  The tombs of Hillat Hamad in Bahri, near the Blue Nile, and attributed to the Sufi Sheikh "Hamad Wad Um Abu Mariouma".
6.  Hillat Khojaly tombs in the city of Khartoum North and attributed to the Sufi Sheikh "Khojali Abdel Rahman".
7.  Al-Jarif cemeteries east and located east of the city of Khartoum.
8.  The tombs of Sheikh Al-Kabashi, north of the city of Bahri, and despite its geographical distance, the burial there acquires a religious dimension.
9.  Ahmed Sharafi tombs in Omdurman, and the majority of its users are from the Ansar al-Mahdi sect.
10. Al-Bakri tombs in Omdurman.
11. The tombs of Sheikh Hamad Al-Neel, west of Omdurman, and their Sufi users.

It is important for us to consider this detail of cemeteries to understand the cultural and social contexts associated with burial, since although there are many cemeteries in urban areas, there seems to be a clear preference among urban residents for burial in specific cemeteries, despite the relatively few burial spaces available.

In theory, anyone in Khartoum city can bury their dead in any of the cemeteries available to them, but in fact that is not true. Based on the information collected during data collection, all burials from families with high income were concentrated in Khartoum in the graveyard Farouk, and Bahri in the graveyard Hillat Hamad, as well as in Omdurman in the graveyards of Al-Bakri and Ahmed Sharafi. In Khartoum, the Burri al-Mahas and Burri al-Lamab cemeteries tended to house the dead of ancient origin. In Omdurman, al-Bakri cemeteries tended to house the dead of ancient origin, and in Khartoum North, Hillat Khojaly cemeteries tended to house the dead of Khartoum North. Political influence in society can also be attributed to the dead. Cemeteries mentioned above are noted for burials. The place of burial of a deceased person is somewhat related to the religious affiliation of the deceased's family. The Ansar al-Mahdi sect, for instance, tends to bury all its dead in the Omdurman cemeteries of Ahmed Sharafi. In the capital of Khartoum, Sheikh Khojaly, Sheikh Hamad, and Sheikh Al-Kabashi, the tombs of the righteous saints of God are also located in the cemeteries belonging to Sufi orders. Among the important observations made here is the fact that Tuti Island, one of the roots in the city of Khartoum, does not have cemeteries. However, because of their religious affiliation with Sheikh Hamad Wad Um Maryouma and Sheikh Khojali Abdul Rahman, they have to endure hardships and transport their dead to these two cemeteries in Khartoum North by dhows.

A lot has been mentioned about how relationships can play a role in burying the dead. It is not common in Sudanese society for a single family to own a cemetery, a practice present in other Arab countries. Here, the kinship factor is the fact that the burial takes place at the gravesite of one's closest relatives, such as one's father or mother, brother, husband, or wife. A religious belief may refer to visiting the deceased and praying for them, or an emotional belief may refer to bringing the family together. As a result, societal concerns about burying the dead where their relatives lived are a factor supporting this trend. In the belief that doing so will make the deceased more comfortable in his or her grave, one wishes to reunite with the deceased with the mother, father, wife, husband, or a child. Despite the high cost of transporting bodies of deceased immigrants or those who died while receiving treatment abroad, this happens in some cases.

Despite the religious interpretation of death for Muslims, this approach to burial is a reassuring and calm interpretation that can be attributed to rational choices. In addition, as the open interviews revealed, the burial next to close relatives is also driven by psychologi-

cal and social factors, such as the importance of symbolism in the neighborhood and the presence of death in the neighborhood's memory (Ahmed 2020).

We note that some families rebury their deceased in cemeteries in an earthen layer in an attempt to highlight their social status, especially in those that are overflowing and lack space for burials. This practice is forbidden by the Sudanese Islamic jurisprudence despite the advisory opinion issued by the Supreme Council on Islamic Jurisprudence (Fatwa Sharia, No. 16, Appendix B).

Depending on the neighborhood, common residence, or daily cooperation, social relationships may influence where the dead are buried. The burying of their deceased in restricted cemeteries can present some complications for residents or outsiders who are ineligible, but a close friendship, work relationship or housing relationship may overcome them. An interview with one of the deceased individuals at Burri al-Mahas cemetery revealed that they were not residents of the area where these cemeteries are, but through the friendship of a neighbor in the Burri area who has extensive social relations, they were able to find a place for burial here.

In light of the cultural and social nuances related to the place of burial of the dead, it is important not to assume it symbolizes the culture of a group as much as the culture of the community. A capital city like Khartoum, representing groups according to political, religious, and sectarian affiliations, is an example of how human groups belonging to different origins can integrate. It seems that the burial place outside the capital is not affected by political, religious, or economic factors based on our observations and experience of society. The place where the dead are buried is also unimportant to a large section of society.

### 6.4. About Burial of the Dead

Mourners gather in a circle around the "funeral yard" of the dead, where one of them loudly declares "Al-Fatihah." Each person raises their hands and reads Surat Al-Fatihah. Afterwards, a place in the cemetery is designated for holding funeral prayers for the dead. Most mourners participate in burying the deceased by burying dirt and bringing the body into the grave. The prayer leader begins the memorial by discussing the exploits of the dead near the grave, thanking those who attended the funeral for comforting the living, and concludes by saying "Fatiha" to the deceased's soul. A common observation regarding burials, especially when the deceased is outside of the country, is that many families tend to break the coffin and then bury the dead instead of burying them inside. (Fatwa Sharia, No. 17833, Appendix C) As a result of local religious beliefs, burial is compulsory within a grave known as "Wad al-Lahd",[2] the locally used term for a grave, as well as psychological, emotional, and sentimental considerations regarding the deceased. The system of communal rooms is unknown in the culture of societal burials. Sudanese communities in the diaspora have experienced many problems as a result. It has been reported on the Sudanese Online website that a dispute has arisen between the Sudanese community in Cairo in the Ain Shams area, which is concerned about the prevention of burial of the dead of Sudanese, and of digging their graves in the way of the deceased. Sudanese diaspora members may be eager to bury their dead in Sudan under the name of "honoring their dead" as shown by this incident. As per the fatwas issued in this regard (Fatwa Sharia, No. 17833), this could also be due to some religious reasons related to the hatred of burying the dead in a coffin.

### 7. Conclusions

In all disciplines of the social sciences, studying death is emotionally and psychologically draining for researchers and scholars. Particularly in Arab societies, there was a noticeable lack of study of death. As a result of its study of death as a socio-cultural situation that is isolated from religious reality, we argue that this research has gone beyond the norm in applied sociology in the Arab region. In addition, the research tends to separate the culture from the group and society by finding social meanings that characterize the group as a whole throughout the stages of societal death. A key finding from the study on the relationship

between death, society, and culture in Khartoum is the re-death within the framework of kinship and neighbourhood to form the social structure. An analysis of social meanings attributed to death found that they were formed from objective and realistic realities of coexistence and were associated with the announcement of death. A number of social and cultural considerations were also taken into account during the burial process, such as the deceased's social, economic, and religious status, and the location of the deceased's family. Rather than representing the culture of the community, the social and cultural dimensions associated with burial of the deceased reflect the culture of the group. Lastly, the research uncovered those intersections between religion and culture regarding death.

**Author Contributions:** Conceptualization, O.S.A. and E.A.; methodology, O.S.A. and A.Z.A.R.; formal analysis A.R.A.A. and E.A.; data curation O.S.A. and A.Z.A.R.; writing—original draft preparation, O.S.A.; writing—review and editing, O.S.A. and E.A. All authors have read and agreed to the published version of the manuscript.

**Funding:** This paper received no external funding.

**Informed Consent Statement:** Informed consent was obtained from all subjects involved in the study.

**Conflicts of Interest:** The authors declare no conflict of interest.

**Appendix A**

A fatwa issued by the General Presidency for Research and Ifta, in the Kingdom of Saudi Arabia, on the subject of using the words (the deceased) and (the forgiven) for referring to a dead person.

Fatwa No. (8217):

Q: I heard some words that some people repeatedly use, so I want to know what is the position of Islam with regard to usage of these? For example: When a certain person dies, people say: (al-Marhoum—The late so-and-so (who received the merciful blessing of Allah), and if he is of a high position, they say: (al-Maghfour lahu—the forgiven So-and-so)). The question is that did they have a glimpse at the al-Lawh al-Mahfooz (the Preserved Tablet) and know that So-and-so was forgiven, and So-and-so has received the mercy of Allah? Therefore, it was urged to question this point, and the Almighty said in his Noble Book: "Remember, O Prophet, when Allah took the covenant of those who were given the Scripture to make it known to people and not hide it,". Please enlighten me.

A: Asserting Allah's forgiveness of a person or His mercy to a person after his death is a matter of the unseen (al-Ghyab) that only Allah the Almighty knows, then whom Allah has informed of that among His angels, messengers, and prophets describing a dead person as being forgiven or had mercy from Allah, by any person other than the angels, messengers and prophets, is not permissible, except for those who are mentioned in a text attributed to the infallible, (prophet Muhammad, peace be upon him), and without that such a description would amount to a mere blind guessing. Allah the Almighty says: (Say, ˹O Prophet˺, None in the heavens and the earth has knowledge of the unseen except Allah, and (He is the Knower of the unseen, disclosing none of it to anyone). It is preferable to ask for forgiveness and mercy from Allah to the dead person instead of assigning the definitive descriptions mentioned in the question. Allah the Almighty says: (Indeed, Allah does not forgive associating others with Him ˹in worship˺, but forgives anything else of whoever He wills. And whoever associates others with Allah has indeed committed a grave sin). In Al-Bukhari's *Sahih al-Bukhari*, (Al-Bukhari Authenticated Collection of the Prophet's hadith), on the authority to Kharijah ibn Zayed Ibn Thabit that (Umm Al-Ala—a woman of the Ansar had pledged allegiance to the Prophet, may Alla's prayers and peace be upon him—told him that the Muhajireen divided a lottery, so Uthman bin

Mazoon flew to us, and we took him into our homes. So fell ill which eventually caused his demise and when he died and was washed and shrouded in his clothes, the Messenger of God, may God bless him and grant him peace, entered into the place, and I said: May Allah's mercy be upon you, Abu al-Sa'ib. I bear witness that you were honored by Allah, He said: How did you know that he was honored by Allah? So, I said: By sacrificing my father, O Messenger of Allah, who will Allah honor then? He said: As for him, what is certainly predestined has befallen him, and by Allah, I wish him well. By the name of Allah, I., though the messenger of Allah, do not know, what He will do to me. She said: By Allah, I will never praise anyone to Allah anymore.)

The Prophet's saying to that woman: (may God's prayers and peace be upon him) (By the name of Allah, I., though the messenger of Allah, do not know, what He will do to me) has happened before he received the revelation that (Indeed, We have granted you a clear triumph ˹O Prophet˺) (so that Allah may forgive you for your past and future shortcomings,) That was before Allah revealed to him that he is one of the people of Paradise.

Good luck. God bless our Prophet Muhammad and his family.

**Appendix B**

A fatwa issued by the Sudanese Islamic Fiqh Academy on the subject of making an earthen layer over graves for burial of more dead people on the same grave.

Date of the fatwa: 7 Dhul Qi'dah 1425 AH corresponding to 18 December 2004

Fatwa Number: 16

The question:

What is the fatwa ruling on filling in old tombs that were filled with a layer of soil at a height of one meter or more, and fixing this layer by heavy machinery, for the purpose of burial of more corpses?

Fatwa:

This action is not permissible due to what has been proven by authentic previous precedents and narrations that the dead are harmed by what is harmful to the living, and that respecting the dead in their graves is like respecting the dead in their homes while they were alive, and because these graves are exclusively allotted to their owners, as their homes and places of visitation, it is not permissible to transgress on them. It is forbidden to trample on them with shoes or to sit on top of graves. Also, there is no necessity or legal need calling for this type of action, because it is possible to allocate another place to bury the newly dead that would satisfy the accepted requirements.

**Appendix C**

What is the Sharia ruling on putting the properly shrouded dead body in an airtight casket or coffin and placing him into the grave?

Fatwa Number No: 17833

Fatwa: According to the Sunnah it is not accepted to bury the dead in a box because that was not done by the Prophet, peace and blessings of Allah be upon him, nor by his companions after him, but it is prescribed to shroud the dead body in three white layers of fabric and place him in a sepulcher in his grave.

**Notes**

[1]    The memorial service is always held at the house of the deceased or the family of the deceased, which is a Sudanese colloquial term. Women and men alike wail when a dear family member dies, this is called the House of Bikaa.

[2]    What is meant by it is the grave, but it is pronounced in the Sudanese colloquial wad al-lahd, and it means the crack that is in the side of the grave, the place of the dead, on the side of the Al-qiblah, and its width is 30 cm and its length is about two meters.

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
