# Peer review of "The Social and Cultural Dimensions Associated with Death in Muslim Communities, a Case Study Khartoum City"

_socsci, doi:10.3390/socsci11090410_

Round 1
Reviewer 1 Report
This article could potentially be a contribution to scholarship given the current lack of research on contemporary Sudanese funerary practices. Unfortunately, it gets bogged down in its presentation of previous scholarship at the expense of any rich contextualized ethnographic detail. More needs to be said about actual Sudanese practices, with detailed examples, and less said on this history of death scholarship in the social sciences. This imbalance is probably because three authors appear to have been involved. They should work closely in the reconceptualizing the revision process.
Additionally, the argument is significantly obscured by incorrect usage and grammar, beginning on the first page and occurring throughout the article. Revision should be attentive to enhancing clarity and logical development of the argument.
One of the article's main claims is that "there is a vacuum in the field of death study" and that "sociology has not paid attention to death" (p. 1). Yet it contains a lengthy discussion of sociological research on the topic on pp. 3-5. Missing from this discussion are several key works, such as Herz's "Death and the Right Hand," Aries' "The Hour of Our Death," not to mention the work of Victor Turner and Gary Laderman. In the Egyptian context, there is the work of Abu Lughod, John Kennedy (Nubian funerary practices), El-Sayyid El-Aswad, and this reviewer. In addition, it gives attention to a random range of topics, including euthanasia, organ transplants, immigration, and terrorism, that don't seem to be connected to the Sudanese context (e.g., pp. 4-5).
Lastly, the footnotes and bibliography need to be reformatted to reflect scholarly conventions (e.g., APA, MLA, Chicago).
Author Response
Thanks for your interest in reviewing the article Please find attached my responses on your suggestions

Reviewer 2 Report
This paper is confusing to read, not only does the English need some work but there are a number of incomplete sentences, repeated sections in random places, and general organizational issues. The main findings and contribution of this research also remain unclear. There are some interesting anthropological accounts here, but the research and findings themselves were thin.
Author Response
Thanks for your interest in reviewing the article Please Below my responses to your suggestions:
- Research questions are clearly designed in the updated version
- The language of the article has been corrected and revised
- The main contribution of this article is to describe the social customs accompanying death in urban society in Sudan
Author Response

(The authors gave the same response as above.)

Round 2
Reviewer 2 Report
This updated manuscript does not show extensive revision addressing previous issues. The paper is difficult/confusing to read (I recommend working closely with an editor to clean it up), and the research is not thoroughly described or interpreted. The topic is interesting, important, and clearly has merit, but the paper still needs extensive work before being publishable.
Author Response
dear
Your recommend has been completed in the attached updated manuscript 2
Thx
Reviewer 3 Report
Accepted in present form
Author Response
thankful for your acceptance
Round 3
Reviewer 2 Report
Huge improvement here! Well done. This has evolved into a solid paper.